# The Low-Cycle Fatigue Behavior, Failure Mechanism and Prediction of SLM Ti-6Al-4V Alloy with Different Heat Treatment Methods

**DOI:** 10.3390/ma14216276

**Published:** 2021-10-21

**Authors:** Jiangjing Xi, Yun Hu, Hui Xing, Yuanfei Han, Haiying Zhang, Jun Jiang, Kamran Nikbin

**Affiliations:** 1Department of Mechanical Engineering, Imperial College London SW7 2AZ, UK; j.xi18@imperial.ac.uk (J.X.); xinghui@sjtu.edu.cn (H.X.); hyuf1@sjtu.edu.cn (Y.H.); 2Aircraft Strength Research Institute, Xi’an 710065, China; haiyingzh13@126.com; 3School of Mechatronics Engineering, Nanchang University, Nanchang 330031, China; 4School of Materials Science and Engineering, Shanghai Jiao Tong University, Shanghai 200240, China

**Keywords:** low-cycle fatigue, titanium alloy, SLM, additive manufacturing, fatigue model

## Abstract

Selective laser melting (SLM) is a promising additive manufacturing (AM) process for high-strength or high-manufacturing-cost metals such as Ti-6Al-4V widely applied in aeronautical industry components with high material waste or complex geometry. However, one of the main challenges of AM parts is the variability in fatigue properties. In this study, standard cyclic fatigue and monotonic tensile testing specimens were fabricated by SLM and subsequently heat treated using the standard heat treatment (HT) or hot isostatic pressing (HIP) methods. All the specimens were post-treated to relieve the residual stress and subsequently machined to the same surface finishing. These specimens were tested in the low-cycle fatigue (LCF) regime. The effects of post-process methods on the failure mechanisms were observed using scanning electron microscopy (SEM) and optical microscopy (OM) characterization methods. While the tensile test results showed that specimens with different post-process treatment methods have similar tensile strength, the LCF test revealed that no significant difference exists between HT and HIP specimens. Based on the results, critical factors influencing the LCF properties are discussed. Furthermore, a microstructure-based multistage fatigue model was employed to predict the LCF life. The results show good agreement with the experiment.

## 1. Introduction

Additive manufacturing (AM) technology can fabricate near-net-shaped parts from the bottom up in a layer-by-layer manner directly from a CAD model without special tooling [1]. This makes AM technology a potential new manufacturing process for components with complex shapes [2]. Additionally, the rapid prototyping techniques enable the production of metallic components, which allows a significant reduction in material consumption compared to the conventional process, especially when manufacturing high-specific-strength metal alloys such as Ti-6Al-4V [2,3]. Due to its high strength, corrosion resistance, and low specific weight, Ti-6Al-4V is ideal for application in aerospace components and biomedical implants.

Massive efforts have been made to develop AM methods [4]. Powder bed fusion technology (PBF) is a specific developed subset of AM technologies which uses a concentrated energy beam to melt a powder bed composed of polymer, metal, or ceramic raw materials layer by layer. Moreover, PBF processes vary on the basis of the type of applied power source, e.g., laser or electron beam. Selective laser melting (SLM), utilizing a laser beam as an energy source, is one of most widely used PBF processes [2], which makes it appealing for the fabrication of Ti-6Al-4V; however, it features drawbacks in terms of the porosity [3], anisotropy result from strongly textured microstructure [4], residual stress [5], and rough surface in as-built conditions [6]. With carefully chosen process parameters, SLM titanium alloys have the possibility to achieve near fully dense parts [7]. Parts with comparable mechanical properties to those of traditionally fabricated titanium alloy have been reported [8,9]. It is evident that, to produce near fully dense products, the key process parameters include the laser power density, layer thickness, sintering rate, and the manufacturing strategy [10,11], which underlie the mechanism of densification during the manufacturing process [11].

In SLM, due to the high cooling rate, there is an anisotropic $\alpha^{\prime }$ martensitic phase contained within prior-β grains oriented epitaxially in a perpendicular direction to the layers [5]. These alloys with a typical microstructure have high strength but poor ductility, as well as a certain anisotropy [12]. Therefore, to a balanced property between strength and ductility, post-treatment processes involving heat have been introduced [8].

One of the main challenges in the widespread adoption of AM technology by industries is the uncertainty in the in-service structural properties of their fabricated parts [13,14]. The main factors contributing to this uncertainty are microstructural heterogeneities and randomly dispersed defects [15]. Consequently, the fatigue performance of AM parts is highly affected. Significant research efforts have been made to investigate the fatigue properties and resistance of AM Ti-6Al-4V. The intrinsic limitations (i.e., thermal history, cooling rate, and cyclic reheating) of AM methods lead to the presence of porosity, surface roughness, and high tensile residual stresses. These restrictions impose a severely detrimental effect on the fatigue performances of as-built components [16]. Among these factors, surface roughness is the most detrimental factor affecting the fatigue performance of AM parts [17]. For internal defects, shorter lifetimes are associated with internal defects near the surface, with inclusion type defects being the worst ones [18]. On the other hand, in actual aerospace or medical applications, rough surfaces are post-machined or polished after fabrication to eliminate the effect of surface roughness. Moreover, residual stress can generally be reduced via heat treatment and surface machining. HIP treatment can eliminate internal defects, decrease the ultimate tensile stress and yield stress, and improve the ductility and fatigue properties [17,18,19,20]. However, the effects of HIP treatment on fatigue performance and its associated fracture mechanisms have not been systematically and quantitatively investigated. Herein, the influencing factors of heat treatment were investigated.

This study investigated the effects of HIP and standard HT methods on the low-cycle fatigue properties of SLM parts. To eliminate the surface roughness effects, which represent the most detrimental factor influencing fatigue property, all surfaces of specimens in this study were machined with identical fine surface finishing. Low-cycle fatigue (LCF) and tensile tests of HT- and HIP-treated samples were conducted to investigate the mechanical properties and LCF performance. The relationships between mechanical properties and LCF properties were discussed and summarized. In addition, this research was aimed at predicting the fatigue life of HT and HIP specimens using the multistage fatigue (MSF) model proposed by McDowell et al. [21] and applied by Torries, B et al. [22] and Ren et al. [23], with the results calibrated for AM Ti-6Al-4V alloys.

## 2. Materials and Methods

### 2.1. Materials and Manufacturing

The samples were additively manufactured by an EOS 280 3D printing system in an argon atmosphere (oxygen < 10 ppm). The details of the SLM process are shown in Figure 1a. The Ti-6Al-4V powder used to produce the samples constituted spherical particles with a size between 13 and 65. The chemical element composition is shown in Table 1. The process parameters were as follows: laser power, 280 W; sintering rate, 1200 mm/s; layer thickness, 40 µm. To avoid negative residual stress during the process and obtain high-density parts, the scanning strategy applied in this manufacturing process was similar to a chessboard. The specimens were manufactured as cylinders and then machined to LCF and tensile specimen geometry, as shown in Figure 1d.

### 2.2. Heat Treatment

After manufacturing, samples for tensile and LCF tests underwent two different heat post-treatments. The first heat treatment method involved heating the samples to 500 °C and holding for 0.5 h, and then holding at 800 °C for 4 h. The second heat treatment method involved hot isostatic pressing at 120 MPa pressure in an argon atmosphere and holding for 2 h. In this study, the heat treatment was chosen to release the residual stress in the SLM parts, whereas the HIP treatment was chosen to close the internal pores. A schematic diagram of the heat treatment methods used in this study can be found in Figure 2.

### 2.3. Tensile and Fatigue Test Methods

Static tensile tests were carried out in a room environment (25 °C, 60% RH) on cylindrical specimens in HT and HIP conditions using a servo-hydraulic universal testing machine INSTRON (Instron, High Wycombe, Buckinghamshire, England) and an axial extensometer (25 mm gauge length, nonlinearity ±0.15% of R.O.). The test was performed in accordance with the ASTM E8/E8M standard at a strain rate of 1.6 × 10^−2^·s^−1^. The yield strength was determined as 0.2% of the offset yield stress. For accuracy and repeatability, two specimens for each manufacturing parameter were evaluated.

To obtain the cyclic stress–strain and the LCF behavior, two types of specimens were investigated using the strain-controlled fatigue test method according to the ASTM E606 standard [24]. The strain-controlled fully reversed uniaxial LCF fatigue test was performed on an INSTRON 8802 testing machine (Instron, High Wycombe, Buckinghamshire, UK) in conjunction with an INSTRON extensometer (2620-601) (Instron, High Wycombe, Buckinghamshire, UK). During the test process, each sample was tested at constant strain amplitude until final failure under a constant strain rate of 1 × 10^−2^·s^−1^. Samples were tested with fully reversed strain (strain ratio R = −1) at four strain amplitude levels from 0.8% to 2.0%. To ensure that the strain rate of all strain level tests was similar and to avoid a detrimental impact on the cyclic behavior, the frequency was adjusted for each strain level. To ensure that the extensometer was tight enough, two rubber rings were used to attach the extensometer’s sharp edges to the specimens. When the specimen was loaded to failure, the reversal life cycles, load response, displacement, and strain values were measured and recorded for each specimen. The experiment loading matrix is presented in Table 2, which includes the applied strain amplitude, Δεa/2, and frequency. Overall, 22 specimens were used (11 for each type); two specimens were used for the largest strain amplitudes, whereas three samples were tested for the intermediate ones. At each specific strain level, three specimens were tested to ensure the reliability of fatigue data. A higher strain level results in smaller scatter; thus, it was acceptable to test only two samples at the highest level.

For both tensile and low-cycle fatigue tests, in all samples, the loading stress axis coincided with the cylinder sample axis. All specimens were polished and machined along the same direction and with the same surface roughness of a maximum of 0.4 µm.

### 2.4. LCF Analysis Model

The strain-controlled LCF approach is a typical method used to estimate crack formation duration in low-cycle conditions. The applied strain to reversal life can be described by the universal approach using the Basquin–Coffin–Manson relationship [22,23,24,25,26,27] given in Equation (1).
(1)$$\Delta \varepsilon /2=\frac{{\sigma^{\prime }}_{f}}{E}{\left(2N_{f}\right)}^{b}+{\varepsilon^{\prime }}_{f}{\left(2N_{f}\right)}^{c},$$
where ${\sigma^{\prime }}_{f}$ and ${\varepsilon^{\prime }}_{f}$ are the fatigue strength coefficient and fatigue ductility coefficient, respectively, b is the fatigue strength exponent, and c is the fatigue ductility exponent. Normally, the right side of Equation (1) represents the elastic and plastic amplitudes, expressed as follows:(2)$$\Delta \varepsilon_{e}/2=\frac{{\sigma^{\prime }}_{f}}{E}{\left(2N_{f}\right)}^{b},$$
(3)$$\Delta \varepsilon_{p}/2={\varepsilon^{\prime }}_{f}{\left(2N_{f}\right)}^{c}.$$

On the basis of the LCF test, the cyclic stress–strain curve can be obtained from the stable hysteresis loop at half of the specimen’s life. The cyclic and static stress–strain curves were fitted using the Ramberg–Osgood equation (Equation (4)) [24].
(4)$$\frac{\Delta \varepsilon }{2}=\frac{\Delta \sigma }{2E}+{\left(\frac{\Delta \sigma }{2K^{\prime }}\right)}^{1/n^{\prime }},$$
where $K^{\prime }$ and $n^{\prime }$ are the cyclic strength coefficient and cyclic strain hardening exponent, respectively, and E  is the elastic modulus.

### 2.5. Microstructure Analysis

To investigate the microstructure, a portion of each specimen was obtained from close to the center of the gauge section. The portions were cut along the longitudinal planes, and the metallographic samples were prepared using a standard mechanical grinding and polishing procedure. The Ti-6Al-4V metallographic specimens were then etched with Kroll’s reagent (2% HF, 6% HNO_3_, and 92% H_2_O). Optical microscopy examination of the microstructure was conducted using a Zeiss Axio optical microscope (Carl Zeiss, Oberkochen, Baden-Württemberg, Germany). Fractography images were conducted using a field-emission scanning electron Microscope (SEM) (Sirion 200, FEI, Hillsboro, OH, USA).

## 3. Experimental Results

### 3.1. Microstructure

The microstructure of as-built SLM Ti-6Al-4V has been reported by many researchers. Because of the high temperature and extremely rapid cooling rate resulting from the additive manufacturing process, a fine needle-shaped martensite $\alpha^{\prime }$ phase was formed [28,29]. The samples exhibited similar $\alpha^{\prime }$ grain and prior-β grain sizes, regardless of the building direction. After 4 h at 800 °C, the fine martensitic microstructure was decomposed into a mixture of α and β. With the effect of heat treatment (800 °C), where the temperature was below the transus temperature for Ti-6Al-4V, the process only relieved the residual stress but did not change the size or morphology of the prior-β grain boundaries. In addition, HT can eliminate the prior-β grain boundary and anisotropy. However, some features of the prior-β grain boundary could be observed, as shown in Figure 3a. The width of prior-β grains was around 100–200 µm. It is noticeable that, on both sides of the prior-β grain boundaries, the α grain colonies were different. It can be observed that the sub-transus temperature treatment decomposed $\alpha^{\prime }$ to α+β, whereby the basket-weave microstructure following HT consisted of a mixture of α and β phases. As shown in Figure 3b, the β phase was located at the α lath grain boundaries, as shown by the dark regions.

With HIP treatment, prior-β grain boundaries were eliminated almost completely. Compared to HT, the length and width of α grains increased significantly, with the average length and width of α laths increasing from 3.7 ± 4 µm and 1.1 ± 0.8 µm to 5.4 ± 6.4 µm and 1.8 ± 1.5 µm, respectively. While the grains of the HIP-treated samples were longer and thicker than HT material, both samples had similar patterns. Heat treatment at a temperature close to the transus point helped to coarsen the α grains and lamellar structure.

More detailed and specific size and microstructure pattern data are shown in Figure 4. The microstructure investigation results indicate no significant difference in sample texture between the two heat treatment methods as the temperatures were all below the β transus. The mixed α and β microstructure of the HIP sample was coarsened compared to the HT sample.

In Figure 5, the 3D map shows the grain morphology and phase distribution of three different sections of the specimens. The microstructure consisted of a mixed α + β phase, whereby lamellar light gray grains represented the α phase, and the β phase was located at the lath boundaries with a dark color [30]. Each specimen at the three sections had similar features in terms of the lamellar grain, but the HIP sample exhibited an increase in α phase and grain size. However, for both heat-treated processes, there was no significant difference in building orientation. It can be inferred that the same process parameters such as laser power, layer thickness, and sintering rate led to a similar microstructure pattern. This indicates that, for these optimized manufacturing parameters, the post-heat process can help reduce the anisotropy of the product.

### 3.2. Tensile and LCF Performance

The tensile properties of SLM Ti-6Al-4V specimens in heat treatment (HT) and hot isostatic pressing (HIP) conditions are presented in Figure 6. Traditionally manufactured wrought specimens were used as a comparison. The results are listed in Table 3. All samples had a similar elastic modulus and ultimate tensile strength. SLM samples in HT and HIP conditions had approximately similar yield stress and ultimate tensile strength, being superior to their wrought counterparts. It can be also observed that HIP treatment helped to improve the ductility according to the elongation results, but there was no significant difference between HT and HIP conditions. The wrought specimens had a balanced property with lower ultimate stress but the best ductility.

For the SLM process, HIP and heat treatment decreased the yield strength and ultimate stress and improved the ductility. This can be ascertained from the test results and literature data [27,28,29,31,32]. From the publications reported, the machined and nonmachined specimens showed no obvious difference in terms of the ultimate tensile property; however, to improve the fatigue properties, the surface roughness needed to be machined to avoid surface defects such as open porosities easily becoming cracks when loaded. Therefore, all specimens in this work were surface-machined using the same standard. In addition, to verify the effect of HIP treatment on closing the internal defects, the density of the samples was measured, and the results are listed in Table 3.

It is commonly recognized that low ductility is a major limitation for additively manufactured *α* + *β* Ti alloys in as-built conditions for both tensile and fatigue verification; therefore, a post-processing heat treatment that can realize stress relief is beneficial for as-built SLM Ti–6Al–4V. The difference between the as-built and heat-treated parts in terms of ductility and elongation is due to the plasticity of the *α* + *β* phases [30]. Figure 6 presents the SEM images of the fracture surface of HT- and HIP-treated static tensile specimens. For all specimens that underwent HT and HIP treatment, the fracture surfaces were generally similar. As presented in the images, the fracture surfaces were rough, and dimples could be observed. The fracture surfaces of round bar specimens consisted of a flat circular region perpendicular to the loading axis and an external annular region close to the surface oriented in the loading direction. The fracture surfaces of heat-treated specimens are illustrated in Figure 7a–c, whereas Figure 7d–f show the HIP-treated specimens. A noticeable difference can be seen in the voids in the flat region of heat-treated specimens. However, the number of pores was limited, indicating good manufacturing quality as a result of the parameters controlling the density of internal defects. No pores were found in the HIP fracture surface, indicating that HIP treatment can help to close the internal defects of AM parts. The size and depth of dimples on both surfaces of specimens following heat treatment and HIP were similar. No significant difference was found in terms of yield stress and ultimate strength between both types of specimens, indicating that ductility highly depends on the heat treatment.

Results of the SLM Ti-6Al-4V low-cycle fatigue testing in this research are summarized in Table 4, which includes the four applied strain amplitudes, $\varepsilon_{a}$, plastic strain, $\varepsilon_{p}$, elastic strain, $\varepsilon_{e}$, stress amplitudes, $\sigma_{a}$, mean stress, $\sigma_{m}$, and fatigue reversals to failure, $2N_{f}$. Figure 8 illustrates the low-cycle fatigue fitting curve of the SLM Ti-6Al-4V samples with two different heat treatments methods. It is noted that the LCF properties are not easy to distinguish between HT and HIP samples. This indicates that the scatter in the LCF data is due to not only the heterogeneous microstructure but also factors such as the phase volume fraction, grain size, orientation, and presence of defects in the samples. The relationships among strain amplitude, elastic strain, plastic strain, and the number of reversals is plotted in Figure 8. The key fatigue parameters are shown in the fitting lines, and the difference between the HT and HIP samples can be compared and identified. According to the fitting lines (Figure 8a,b), the life trends of both samples were between 0.8% and 1.2%.

To investigate the mechanism of LCF, one typical sample conducted under a strain amplitude of 1.2% was picked, and the results are depicted in Figure 9. Before the steady-state stress stage, there was an initial cyclic softening process (Figure 9a). The plastic strain evolution process showed a similar trend, whereby, after the softening stage, its plastic strain increased gradually throughout the process (Figure 9b). In addition, the stress–time trend in Figure 9c agrees with the stress–N _f_ curves. The stress–time relationship in the first 100 s (orange dashed rectangle in Figure 9c) is shown in Figure 9d, showing that the maximum and minimum values on the curve decreased with time. Furthermore, the stress response showed an asymmetrical behavior, whereby the maximum tensile stress was 1050 MPa and the minimum compression stress was −1130 MPa, indicating the specific material response under loading conditions. In addition, the loading and unloading elastic modulus decreased over cycles throughout the process.

### 3.3. Cyclic Softening Behaviours and LCF Performance

Cyclic softening or cyclic hardening is a general behavior when alloys are subjected to cyclic load. The cyclic softening phenomenon can be seen in Figure 9a–c, shown that the stress feedback decreased when samples were subjected to cyclic strain-controlled load. The detailed cyclic softening behavior of both types of samples at specific cycles under four strain amplitudes (0.8%, 1.0%, 1.2%, and 2.0%) is shown in Figure 10. The stress trend of both types of samples agrees with that of the typical sample (Figure 8). The sizes of the hysteresis loops of both types of samples increased with the increase in strain amplitude from 0.8% to 2.0%. Moreover, the HT and HIP samples had similar behavior [33]. Generally, cyclic softening or hardening behavior is relevant to the initial state of the material [34,35,36,37,38,39,40].

Furthermore, Figure 11 compares the stable hysteresis loops of the 10th cycle and half-life cycle under all strain amplitudes of both types of samples. The increase in strain resulted in an increase in the size of hysteresis loops. The maximum stress at half-life was reduced compared to that at the 10th cycle. The hysteresis loop behavior corresponded to the cyclic softening ratio. To investigate the fatigue behavior, the stress and plastic response to $N{\;}_{f}$ was plotted and analyzed (Figure 12). At multiple strain amplitudes, the maximum stress decreased while the plastic strain increased. This also demonstrates the cyclic softening behavior of the SLM Ti-6Al-4V samples. It is noted that, with the increase in strain amplitude, there was an increase in plastic strain amplitude ($\Delta \varepsilon_{a}/2\;\;$< 1.0%). The plastic strain increased throughout the loading process, with damage occurring at an extreme value, as indicated by a sharp jump.

### 3.4. Fractographic Analysis

To investigate the fatigue crack initiation and crack propagation mechanism, fractography analyses of both types of samples at four strain amplitudes (0.8%, 1.0%, 1.2%, and 2.0%) were performed. For all samples investigated at different strain levels, the fatigue crack initiation cites were located at near the surface or subsurface. All cracks initiated from the subsurface region and exhibited a flat feature, and the final fracture area occurred at an approximate 45° angle. The fracture surface had a different appearance at various strain amplitudes. As the strain amplitudes increased, the angle of the fracture surface and loading axis increased in both types of samples (shown in Figure 13). In addition, the area of the steep region increased as the strain level increased. The fracture surface of the HT sample loaded at 2.0% showed a steep failure surface without any feature of fatigue cyclic loading. These results reveal that the strain amplitude had a significant effect on the fracture morphology of HT- and HIP-treated samples.

A fractography investigation of both types of SLM Ti-6Al-4V at four strain amplitudes was conducted to determine the effects of the heat treatment methods on crack initiation and propagation. It can be observed that the fracture features were similar at the specific strain amplitudes for both types of samples (Figure 14). For instance, fatigue striation can be found in the crack propagation area at lower strain amplitude (0.8%, 1.0%, and 1.2%), whereas it was scarce at the higher load level (2.0%). This suggests that the low-cycle fatigue of the samples exhibited different crack propagation performance at different strain amplitudes.

In addition, it was found that there was more than one crack initiation location for both types of samples, and no defect was found close to the crack initiation. A typical fatigue fracture surface of HIP samples at 1.0% strain amplitude was investigated (Figure 15). Similar to typical fatigue fractography, the crack initiated from the surface, and striations could be found in the crack growth area (Figure 15a,b); the propagation steps were also found (Figure 15c,d).

Furthermore, the defects in the fracture surface were investigated by SEM to identify their size and location (see Figure 16); the sizes are shown in Table 5. It is noted that no defects were found in the fracture surface of HIP samples, and no defects were found in critical regions such as the surface and subsurface where cracks initiate under normal circumstances. These findings indicate that the process of HIP can significantly improve the internal defects, and they prove indirectly that the defects in the surface and subsurface play a key role in affecting the fatigue life of SLM Ti-6Al-4V. This agrees with the results reported in [19,36], stating that the surface of AM samples must be improved to reduce surface roughness, thereby avoiding surface defects that lead to crack initiation.

## 4. Discussion

### 4.1. Investigation of AM Ti-6Al-4V LCF Properties

Generally, the fatigue properties of materials manufactured by AM Ti-6Al-4V are significantly dependent on the microstructure, as well as internal defects. To identify the critical factors influencing LCF properties, LCF data were selected from the literature [23,35,36,37,38,39,40,41] for comparison to investigate the relationships between the basic mechanical properties and LCF properties (see Table 6). Two types of samples post-processed by heat treatment and hot isostatic pressing, which led to different internal defects and microstructure features, were researched in this work. The fatigue failure cycles were scattered at each strain amplitude from 0.8% to 2.0%. It is normal that low-cycle fatigue life represents large scattering at a given strain amplitude [42]. To perform a comparison of experimental data using various manufacturing parameters, a unified formula was applied. The Coffin–Manson equation is a proper fitting method that reflects the LCF properties and basic characteristics of different datasets. Therefore, to compare the LCF property data of samples with different parameters, the relationship curve of $\Delta \varepsilon /2–2N_{f}$ was used and fitted [43]. The results indicate that a higher strain amplitude led to fewer fatigue cycles; there were <100 fatigue life cycles when Δε/2 > 2.0%, whereas there were typically >104 cycles at strain amplitudes lower than 0.6% (see Figure 17a). It can be noted from the fitting curve that the LCF performance of wrought Ti-6Al-4V samples was superior to most AM counterparts. It is noticeable that the HT- and HIP-treated samples in this work had better LCF properties at strain amplitudes Δε/2 < 0.8% than their AM counterparts, even wrought ones.

Furthermore, the fatigue of HT samples was slightly higher than that of HIP-treated samples at high strain amplitude, with the gap between both turns increasing with a decrease in strain amplitude. According to the reference data, yield stress plays an important role in LCF properties. Specimens with high yield stress have superior LCF properties, especially at a very-low-cycle fatigue stage; this result can be verified by the data of SLM Ti-6Al-4V ELI (Figure 17a). However, ductility in the form of tensile elongation also plays a major role in LCF properties. SLM Ti-6Al-4V ELI had the highest yield stress (~1015 MPa), but low ductility (~10%), resulting in inferior LCF life at low strain amplitude. In addition, we found that the HT SLM and HIP SLM samples in this research had similar ductility (~17.9% and ~19%) and similar yield stress (~964 MPa and ~913 MPa), resulting in similar fitting curves; clothe slight decrease in yield stress led to superior LCF properties. More evidence of this performance can be seen for the HIP TC4 (ductility ~12.3% and yield stress 872 MPa) and as-built Ti-6Al-4V (ductility ~11% and yield stress 893 MPa). The same performance can also be verified by the as-built DLD (YS 908, EL 3.8%) and HT DLD (YS 957, EL 3.4%) data. Therefore, higher yield stress results in an increase in LCF life at high strain amplitude, and an increase in ductility leads to an increase in LCF life at low amplitudes. These results may have been caused by the HT SLM samples having a finer α-phase (average size 4.93 ± 1 μm) than the HIP-treated SLM samples with a coarser α-phase (average size 8.86 ± 1 μm); these microstructure features can be observed in Figure 3; Figure 4. It can also be concluded that the heat treatment process is essential for an improvement of the fatigue properties of AM Ti-6Al-4V materials.

It is commonly regarded that fatigue life cycles lower than 105 cycles [44] are considered “low-cycle fatigue”. Considering this threshold, the relationship of low-cycle fatigue properties, yield stress (YS), and elongation to failure (EL) can be expressed as a relational graph (Figure 18). The improvement of yield stress contributes to an increase in LCF properties at lower strain amplitude, while an increase in ductility enhances the LCF life at high strain amplitude. Therefore, increasing both YS and EL improves the LCF properties at all strain amplitudes. According to the relationship of these material characteristics and behavior, it can be predicted that HIP-treated SLM samples would have a higher fatigue life at lower strain amplitude and better ductility, thus enhancing the HCF properties.

It is generally regarded that the fatigue properties of additive manufacturing Ti-6Al-4V are also influenced by the microstructural features [3,45], which can be formed through different post-processes. For AM samples, proper heat treatment methods make it possible to obtain fatigue properties similar to wrought counterparts. Compared to traditionally wrought Ti-6Al-4V with equixially mixed α and β phase microstructural features, AM samples have a more complex microstructure including an acicular $\alpha^{\prime }$ phase and large columnar grains, which result in anisotropic mechanical properties. From the microstructural observation of the AM samples, similar LCF properties may be obtained with different grain sizes and texture characteristics; therefore, there is a balance of the tensile strength and elongation properties. To obtain excellent properties, a proper combination of equiaxial and lamellar α phases is required. It can be seen from the optical results that HIP samples have a coarsened grain size compared to HT samples, because the pressure and high-temperature conditions result in an increased grain size of HIP samples, while the high temperature helped to relieve the residual stress and improve the ductility of both types of samples. However, the heat treatment samples retained the tensile strength with finer acicular grains and a balanced plasticity, which is why HT samples had a superior performance in the LCF test.

### 4.2. The Difference in Failure Mechanism of HT and HIP SLM Ti-6Al-4V

In general, parts manufactured by SLM or any other power-based AM techniques are prone to porosity. Internal AM defects such as porosity led to a negative impact on the specimen fatigue properties [46,47]. Pore shape, size, location, and amount will impact the fatigue behavior of AM parts. Bigger, more irregular pores that are closer to the outer surface will affect the fatigue life more detrimentally [41]. Due to optimized manufacturing parameters and heat treatment methods, limited internal pores were found on the fracture surface, thus preventing a dominant effect on fatigue life. In addition, HIP treatment helped to close the internal defects detected by SEM. The effect of defects on the fatigue crack behavior are investigated and summarized in Figure 19.

In general, the fatigue surface includes the fatigue crack initiation and propagation regions, as well as the final shear fracture region. In this research, rather than relying on initiation from pores, it was found that cracks initiated at alpha colony boundaries in the region close to the outer surface of the samples. At the initiation stage, microcracks prefer propagating along α-laths within the mixed ($\alpha_{s}+\beta$) microstructure [41,47]. HT or HIP α+β titanium alloys with lamellar microstructures have similar crack initiation mechanisms. In addition, as reported in [48], cracks in AM parts are normally generated from α colonies with larger grain sizes. Compared to HT samples, microcracks were easily originated on the HIP sample surface. The crack initiation sites and crack propagation areas are shown in Figure 20. Figure 20a (HT) and Figure 20d (HIP) show the three-dimensional scanned fracture surfaces, which could be divided into three areas. The first is the crack initiation area, while the second is the propagation region. Some ridges due to the effect of high stress can be observed on the fracture plane (shown in Figure 20a,d). Compared to the HIP sample, numerous paralleled branch secondary cracks could be observed on the HT sample fracture plane. This phenomenon can be regarded as one reason why HT samples had a superior LCF life to HIP samples, because the energy was dispersed to multibranch cracks, and the crack propagation trend could be retardant [29].

With the increase in crack propagation, the fatigue cracks can penetrate α-laths; the intragranular crack propagation can be observed in Figure 20c (HT) and Figure 20f (HIP). Cracks that propagated along the *α* + *β* grain lath boundaries can be also observed in the HT crack path (Figure 20c). It is shown that crack tip turned and was deflected more frequently within the finer microstructure of the HT sample, leading to a zig-zag crack path and a rougher fracture surface on the micro and mesoscale.

It is generally believed that the lack of ductility in AM samples contributes to a shorter fatigue life, especially at high strains such as in the LCF test. In this study, there was no obvious difference between the two heat treatment methods and, therefore, the predominant factor influencing the LCF life was the microstructural features.

### 4.3. MSF Fatigue Life Prediction Model Calibration

McDowell et al. [21] proposed and developed a fatigue prediction model named the microstructure-based multistage fatigue (MSF) model, which was applied successfully in aluminum alloy 7075-T651 [49], as well as materials such as AISI 316 stainless steel [50]. This model was also successfully utilized for laser AM Ti-6Al-4V by Torries et al. [22] and calibrated for high-power laser-directed energy deposited Ti-6Al-4V in [23] by Ren et al. This microstructurally sensitive fatigue model is difficult to calibration but very suitable for AM components which do not have uniform microstructural characters and defects due to the unpredictable thermal process during manufacture [14]. The model can be extended to both low-cycle fatigue (LCF) and high-cycle fatigue (HCF). The MSF model takes various stages of fatigue damage evolution determined by experiment into consideration, including crack incubation and microstructural crack (MSC), physically small crack (PSC), and long crack (LC) growth. In this model, the total life of fatigue$N_{Total}$ is expressed as
(5)$$N_{Total}=N_{Inc}+N_{MSC}+N_{PSC}+N_{LC}=N_{Inc}+N_{MSC/PSC}+N_{LC},$$
where $N_{Inc}$ is the number of cycles to incubate a crack, *N_MSC_* is the number of cycles for the growth of an MSC, $N_{PSC}$ represents the life cycles for the growth of a PSC, and $N_{LC}$ represents the life cycles for long crack growth to final failure. For SLM Ti-6Al-4V, *N_LC_* is not observed and can be neglected [22,23].

In general, $N_{Inc}$ is evaluated using the parameter β in a modified Coffin–Manson law at the microscale.
(6)$$C_{Inc}N_{Inc}^{\alpha }=\beta ,$$
where β represents the nonlocal damage parameter surrounding the inclusion in the modified Coffin–Manson law for incubation, and $C_{Inc}$ and α denote the linear and exponential coefficients. The damage parameter can be estimated as follows:(7)$$C_{Inc}=C_{n}+\frac{1}{0.7}\langle \frac{l}{D}-0.3\rangle \left(C_{m}-C_{n}\right),$$
(8)$$\beta =\frac{\Delta \gamma_{\max}^{P^{*}}}{2}=Y{\left(\varepsilon_{a}-\varepsilon_{th}\right)}^{q},\frac{l}{D}<\eta_{\lim},$$
(9)$$\beta =\frac{\Delta \gamma_{\max}^{P^{*}}}{2}=Y(1+\xi \frac{l}{D}){\left(\varepsilon_{a}+\varepsilon_{th}\right)}^{q},\frac{l}{D}>\eta_{\lim},$$
(10)$$Y=[y_{1}+(1+R)y_{2}]{\left(\frac{D_{0}}{D}\right)}^{d^{\prime }},$$
(11)$$\frac{l}{D}=\eta_{\lim}\frac{\langle \varepsilon_{a}-\varepsilon_{th}\rangle }{\varepsilon_{per}-\varepsilon_{th}},\frac{l}{D}<\eta_{\lim},$$
(12)$$\frac{l}{D}=1-(1-\eta_{\lim}){\left(\frac{\varepsilon_{per}}{\varepsilon_{a}}\right)}^{r},\frac{l}{D}>\eta_{\lim},$$
where $C_{n}$ is the nucleation and small crack growth at the inclusion coefficient in the HCF regime, $C_{m}$ is the Coffin–Manson coefficient for incubation, $\varepsilon_{a}$ is the remotely applied strain amplitude, and $\varepsilon_{th}=0.29S_{ut}/(E(1-R))$ and $\varepsilon_{per}=(0.7\sim 0.9)\sigma_{y}^{cyc}/E$ are the strain threshold for damage incubation and the percolation limit for microplasticity, respectively [49]. The geometric parameters include D, l/D, and ξ, which are the size of the pertinent inclusion at which fatigue is incubated, the nominal ratio between the plastic zone size and the diameter of the fatigue damage, and the geometric factor determined in micromechanical simulations, respectively. In the relationship of remote strain and local plastic shear strain, q is the exponent. $\eta_{\lim}$ is the limiting factor that defines the transition from constrained microplasticity to unconstrained microplasticity at the notch, and r is a shape constant for the transition to limit plasticity. $D_{0}$ is the diameter of the pore developed at the crack initiation site, and $d^{\prime }$ is the effect of the pore size on local plastic strain. Constants $y_{1}$ and $y_{2}$ are correlated with the parameter Y. The function is expressed as <f>=f if f≥0; otherwise, <f>=0.

The MSC/PSC growth law is dominated by the range of crack-tip displacement, ΔCTD, which is proportional to crack length, as expressed below.
(13)$$N_{MSC/\mathrm{PSC}}=\frac{1}{\chi \Psi }\ln\left(\frac{\Psi a_{f}+\Omega -\Delta CTD_{th}}{\Psi a_{i}+\Omega -\Delta CTD_{th}}\right),$$
where $\Psi =f(\bar{\varphi })C_{\mathrm{II}}\left(\frac{DCS}{DCS_{0}}\right){\left[\frac{U\Delta \hat{\sigma }}{S_{ut}}\right]}^{n}$, $\Omega =C_{Ι}\left(\frac{DCS}{DCS_{0}}\right){\left(\frac{\Delta \gamma_{\max}^{P}}{2}{|}_{macro}\right)}^{2}$, where $C_{Ι}$ and $C_{\mathrm{II}}$ are the LCF and HCF coefficient, and *n* represents the exponents for the HCF regime. The threshold value for crack-tip displacement in an HCP titanium matrix is $\Delta CTD_{th}=2.95\times 10^{-4}\mu m$ [51]. The equivalent applied stress range ${\bar{\sigma }}_{a}$ is ${\bar{\sigma }}_{a}=\left[\frac{3}{2}\frac{\Delta \sigma_{ij}^{\prime }}{2}\frac{\Delta \sigma_{ij}^{\prime }}{2}\right]$, and the maximum principal stress range $\Delta \sigma_{1}$ is $\Delta \hat{\sigma }=2\theta {\bar{\sigma }}_{a}+(1-\theta )\Delta \sigma_{1}$. The parameter θ is in the range of 0≤θ≤1, where θ=1 means that von Mises stress is dominant, while θ=0 means that maximum principal stress is dominant. Moreover, χ is constant for a specific microstructure. DCS and $DCS_{0}$ represent the dendrite cell size and reference dendrite cell size, respectively. The parameter U represents the load ratio parameter in the form of $U=\frac{1}{1-R}$ for R≤0 and U=1 for R>0. ${\left.\frac{\Delta \gamma_{\max}^{p}}{2}\right|}_{macro}$ is the macroscopic maximum plastic shear strain amplitude, which can be directly calculated using the remote applied strain amplitude $\left(=\frac{3}{2}\frac{\Delta \varepsilon^{p}}{2}=\frac{3}{2}{\left(\frac{\Delta \sigma }{2K^{\prime }}\right)}^{\frac{1}{n^{\prime }}}\right)$ under cyclic loading conditions. The function $f(\bar{\varphi })$ presents the microporosity level $\bar{\varphi }$. More detailed descriptions and explanations of the MSF model were reported in [45,46,47,48,49,50,51].

Among the MSF model parameters, the constants and coefficients in Equations (9)–(13) for MSF prediction were calculated with reference to the experiment results herein and in [44,45,46,47,48,49,50]. Some material parameters were picked from studies by Torries et al. [22] and Ren [23] with similar manufacturing methods and microstructural features. According to the equations and parameters, the incubation life and MSC/PSC life can be predicted. The upper and lower bounds of the MSF life can also be obtained.

As shown in Figure 21, the fatigue life prediction results of the two heat-treatment processes obtained using the MSF model were plotted. Figure 21a,c exhibit the total fatigue life and the fractions, incubation life, and MSC/PSC life of HT and HIP samples, respectively. Figure 21b,d present the upper and lower bounds of the fatigue life according to experimental data; here, we assumed that the cracks formed and initiated at casting pores of diameter 5 and 500 µm, respectively. Taking the dispersity and the estimation of parameters in the MSF model into consideration, the prediction fatigue life curves showed agreement with the experiment results. However, at a high strain amplitude level ($\Delta \varepsilon_{a}/2$ = 2%), there was a noticeable error between the experimental data and prediction. This is because only one sample was evaluated at high strain amplitude, leading to more unpredictable low-cycle fatigue results. Therefore, the MSF prediction model was deemed acceptable in this research. The upper bound and prediction life adequately fit the experiment data, as we assumed the smallest and largest incubation pore diameters as 5 and 500 µm; thus, actual internal defects or voids of size smaller than 80 µm could have similar effects on the prediction results (Table 7).

Compared to Ren’s [23] study, this study showed a better agreement between predicted and experimental results at a high strain level. The results fit well when the loading strain level decreased for both HT and HIP samples. The error between the test and model calibration results was noticeable, potentially due to only one sample being evaluated, as well as errors were generated from scattering. Thus, fatigue life can be more accurately predicted following further data collection and the investigation of additional parameters.

## 5. Conclusions

In this study, the low-cycle fatigue (LCF) and tensile properties of Ti-6Al-4V alloy fabricated by SLM using the standard HT and HIP methods were investigated and compared to those of other AM technologies. The major conclusions can be summarized as follows:The static tensile properties of HT- and HIP-treated SLM Ti-6Al-4V were similar and comparable to their wrought counterparts.The HT- and HIP-treated SLM Ti-6Al-4V alloys showed higher LCF lives than most AM Ti-6Al-4V samples in the literature and even better LCF properties than the standard wrought samples at lower strain amplitudes.The material underwent obvious cyclic strain-softening behavior at all strain levels from 0.8% to 2.0%. When the strain amplitudes increased, the cyclic softening degree also increased.The hot isostatic press (HIP) process helps to close the internal defects, and almost no defects were found in the HIP-treated samples. Defects were found on the HT sample fracture surface, indicating that defects were not the dominant factor influencing the LCF properties of SLM Ti-6Al-4V; furthermore, the interior defects were not significantly detrimental to LCF performance.Multiple crack initiation sites were found for both HT- and HIP-treated samples. All crack origins were near the surface, and none of them were pores or voids.The behaviors of fatigue crack initiation and propagation were investigated for the HT- and HIP-treated samples. In terms of the crack initiation region, HT samples had more branched secondary cracks than HIP-treated samples, and cracks propagated along α + β lath boundaries. The crack paths of HT samples exhibited a more zig-zag pattern than their HIP counterparts. As the crack propagated, intragranular fatigue fracture became the main crack growth phenomenon.The LCF prediction life curve of HT- and HIP-treated SLM samples obtained using the microstructure-based multistage fatigue (MSF) model showed good agreement with the experimental results.

## Figures and Tables

**Figure 1 materials-14-06276-f001:**
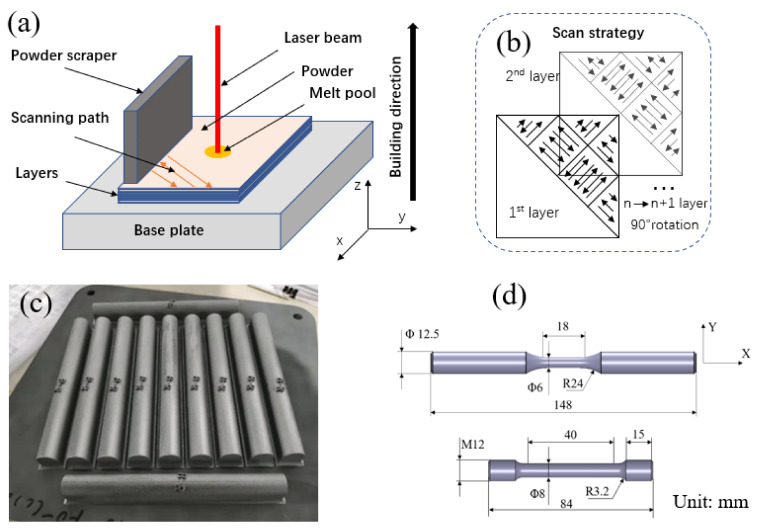
Schematic of selective laser melting Ti-6Al-4V: (**a**) schematic of part forming; (**b**) scan strategy; (**c**) manufactured cylindrical bar; (**d**) final geometry of LCF and tensile specimens.

**Figure 2 materials-14-06276-f002:**
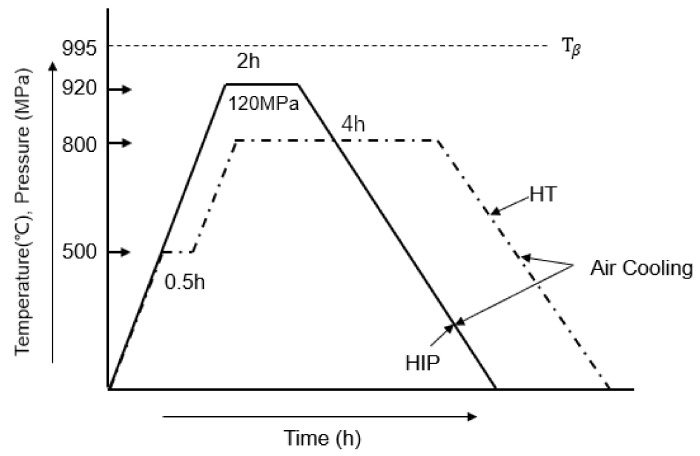
Schematic diagram of the heat treatment methods used.

**Figure 3 materials-14-06276-f003:**
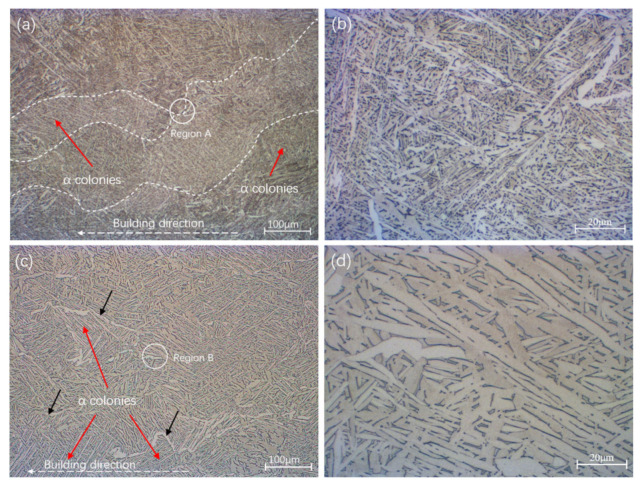
Typical microstructure of the SLM Ti-6Al-4V SLM samples in HT and HIP conditions: (**a**) heat-treated samples; (**b**) enlarged image of region A; (**c**) HIP-treated samples; (**d**) enlarged image of region B. The α phase is represented by light regions, while the β phase is represented by dark regions. The white dashed line is the prior-β grain boundary, while black arrows indicate the α grain boundary.

**Figure 4 materials-14-06276-f004:**
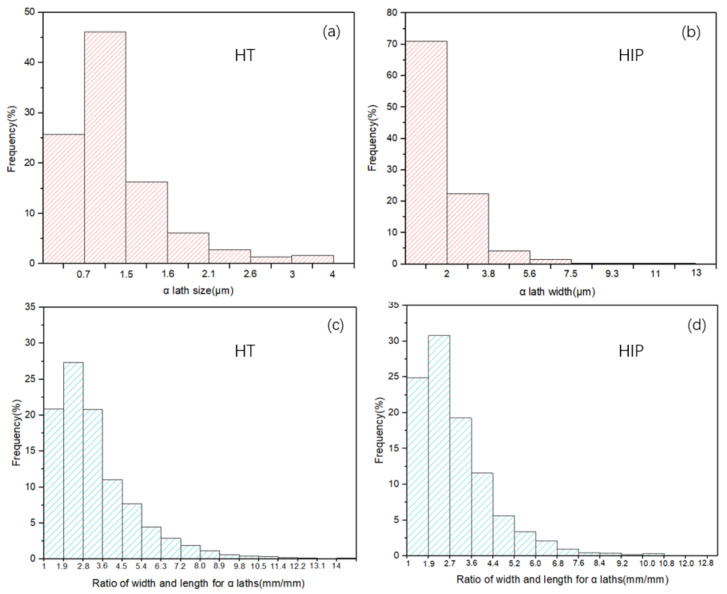
(**a**,**b**) α-lath sizes of HT-treated samples and HIP-treated samples, respectively, (**c**,**d**) Ratio of width and length for HT and HIP sample α-laths.

**Figure 5 materials-14-06276-f005:**
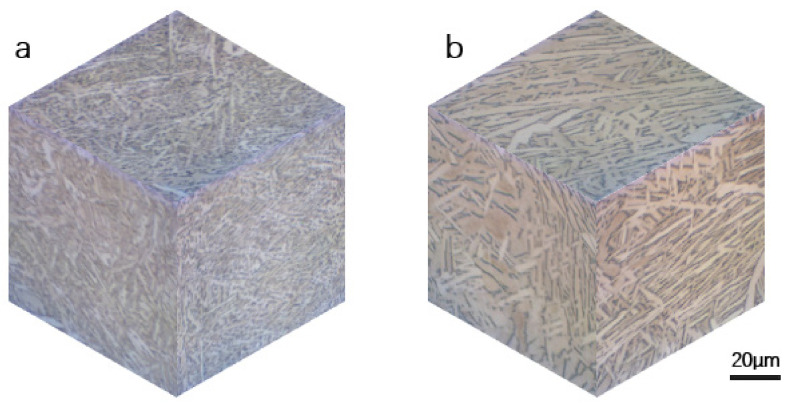
Three-dimensional microstructure map of SLM Ti-6Al-4V SLM samples: (**a**) heat-treated sample; (**b**) HIP-treated sample.

**Figure 6 materials-14-06276-f006:**
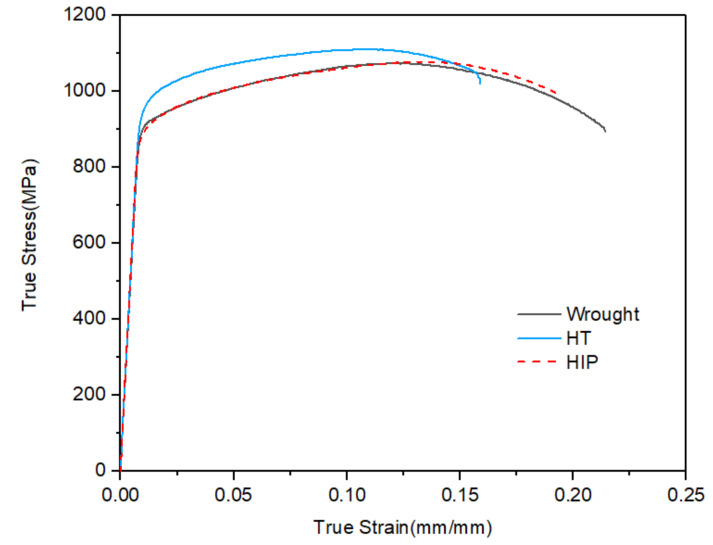
Tensile stress–strain curve for wrought and two types of AM-machined specimens under tension loading.

**Figure 7 materials-14-06276-f007:**
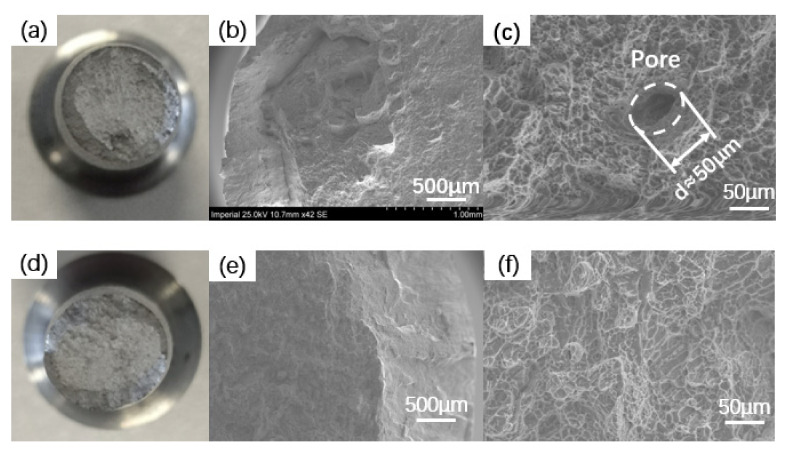
Fracture surface of tensile specimens (HT- and HIP-treated): (**a**–**c**) tensile fracture surface of heat-treated specimen; (**d**–**f**) tensile fracture surface of HIP-treated specimen.

**Figure 8 materials-14-06276-f008:**
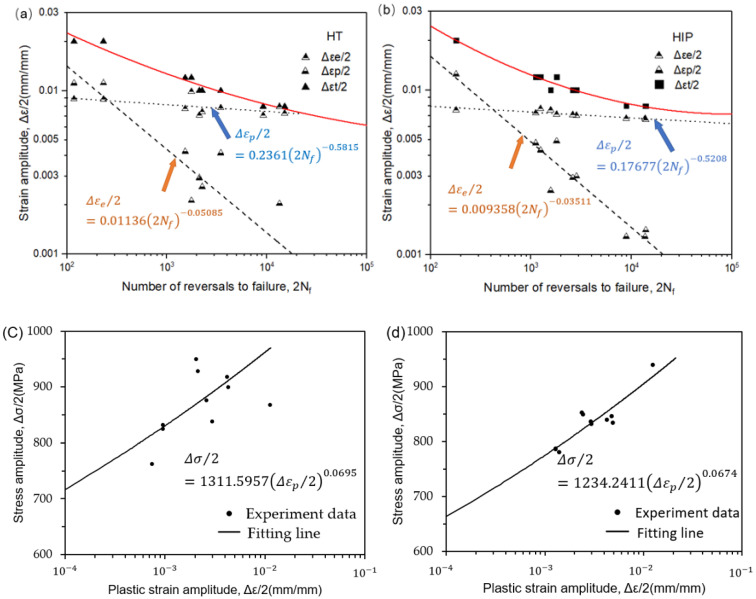
Low-cycle fatigue (LCF) properties for the samples of SLM Ti-6Al-4V: (**a**,**b**) LCF data of heat-treated and HIP-treated samples; (**c**,**d**) stress amplitude and plastic amplitude of both types of samples.

**Figure 9 materials-14-06276-f009:**
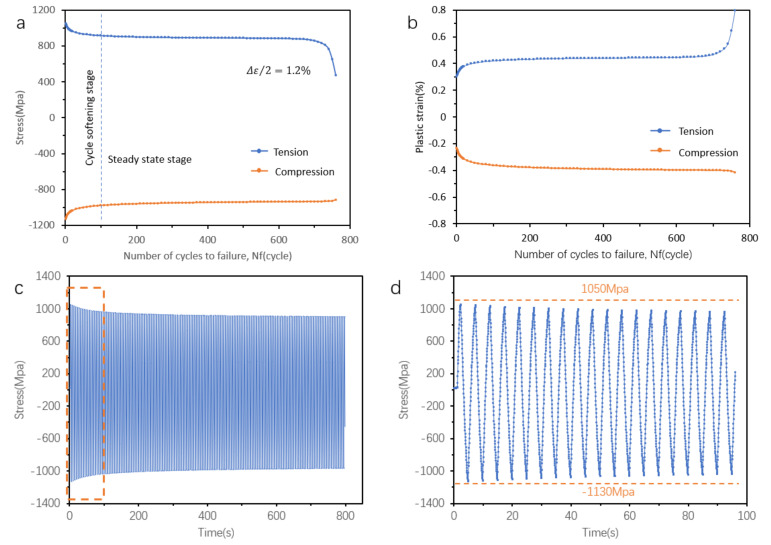
Fatigue process of the HT samples of SLM Ti-6Al-4V: (**a**) stress evolution over cycles to failure; (**b**) plastic–strain over cycles to failure; (**c**,**d**) stress response of the whole test process and in the initial cycles.

**Figure 10 materials-14-06276-f010:**
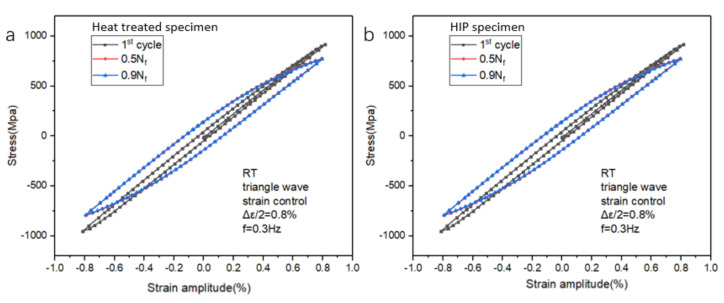
Cyclic softening behavior of HT- and HIP-treated samples at (**a**,**b**)$\;\Delta \varepsilon_{a}/2$ = 0.8%, (**c**,**d**) $\Delta \varepsilon_{a}/2$ = 1.0%, (**e**,**f**) $\Delta \varepsilon_{a}/2$ = 1.2%, and (**g**,**h**) $\Delta \varepsilon_{a}/2$ = 2.0%.

**Figure 11 materials-14-06276-f011:**
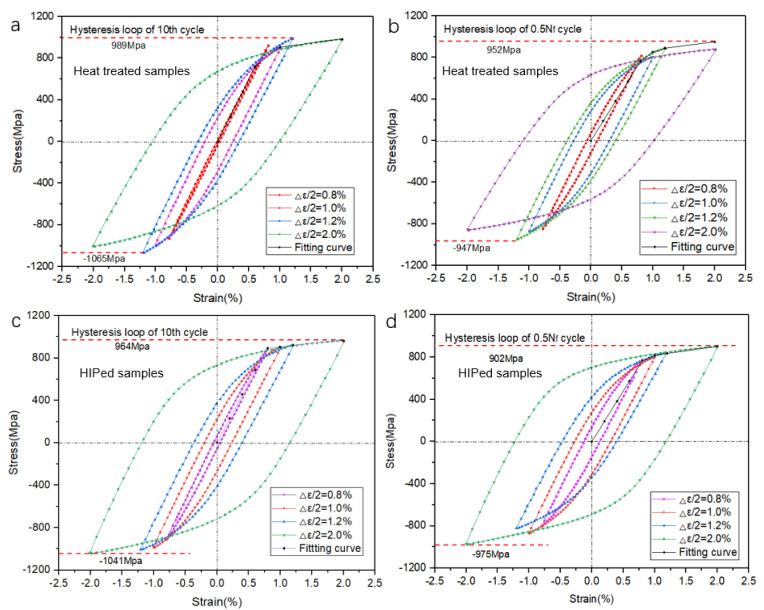
Hysteresis loops of SLM Ti-6Al-4V samples with HT and HIP treatments under different strain amplitude: (**a**) 10th cycles of HT samples; (**b**) half-life cycles of HT samples; (**c**) 10th cycles of HIP-treated samples; (**d**) half-life cycles of HIP-treated samples.

**Figure 12 materials-14-06276-f012:**
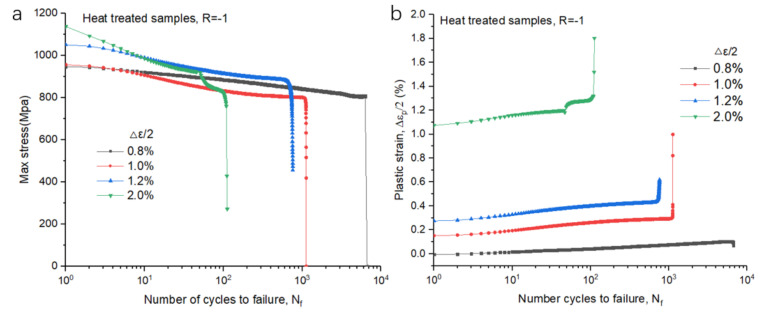
Maximum stress and average plastic strain amplitude and $N_{f}$ at given strain amplitude of HT- and HIP-treated samples under different strain amplitude: (**a**) maximum stress as a function of number of cycles to failure in HT samples; (**b**) plastic strain amplitude and *N_f_* of HT samples; (**c**) maximum stress as a function of number of cycles to failure in HIP-treated samples; (**d**) plastic strain amplitude and *N_f_* of HIP-treated samples.

**Figure 13 materials-14-06276-f013:**
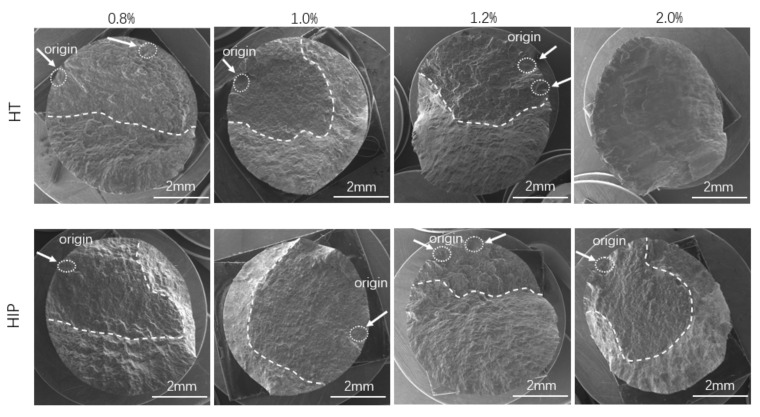
LCF fractography of HT- and HIP-treated samples at different strain amplitudes. The dashed circles and dashed lines separate the crack growth regions of the fracture surface at different stages under corresponding strain. The strain amplitudes were 0.8%, 1.0%, 1.2%, and 2.0% (from left to right).

**Figure 14 materials-14-06276-f014:**
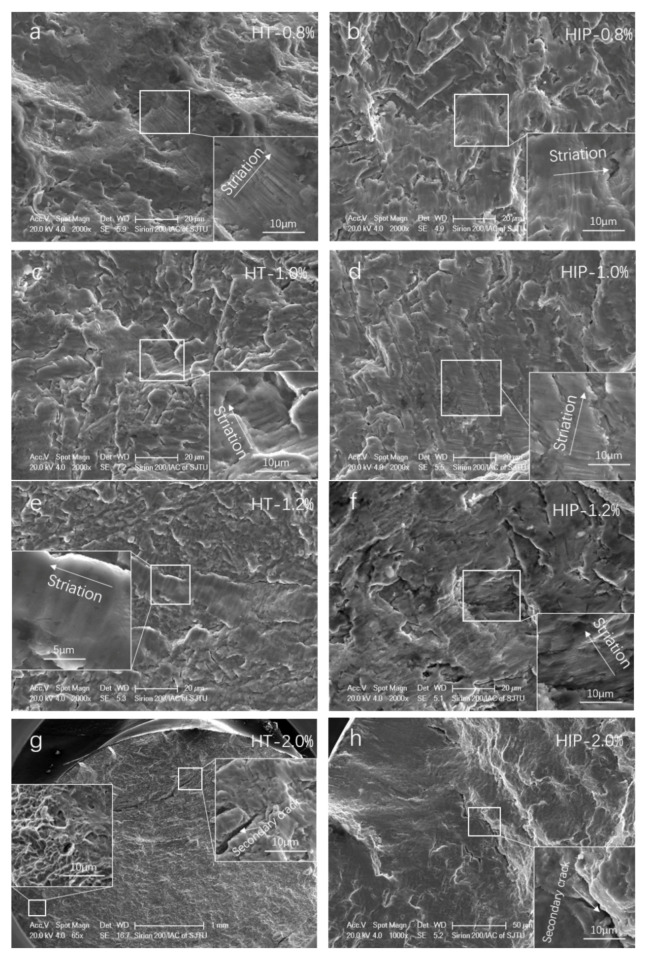
Low-cycle fatigue fracture surface and crack propagation features of HT- and HIP-treated samples: (**a**) HT at 0.8% strain amplitude; (**b**) HIP at 0.8% strain amplitude; (**c**) HT at 1.0% strain amplitude; (**d**) HIP at 1.0% strain amplitude; (**e**) HT at 1.2% strain amplitude; (**f**) HIP at 1.2% strain amplitude; (**g**) HT at 2.0% strain amplitude; (**h**) HIP at 2.0% strain amplitude.

**Figure 15 materials-14-06276-f015:**
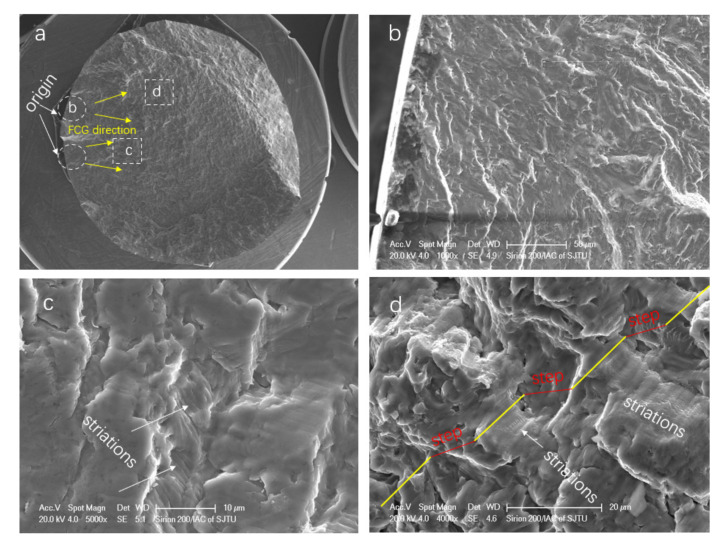
Typical low-cycle fatigue fracture surface features of HIP-treated sample at 1.0% strain amplitude: (**a**) morphology of a low-cycle fatigue specimen (FCG denotes crack propagation); (**b**) crack initiation region; (**c**) striations in the crack growth region; (**d**) transition region with striations and steps.

**Figure 16 materials-14-06276-f016:**
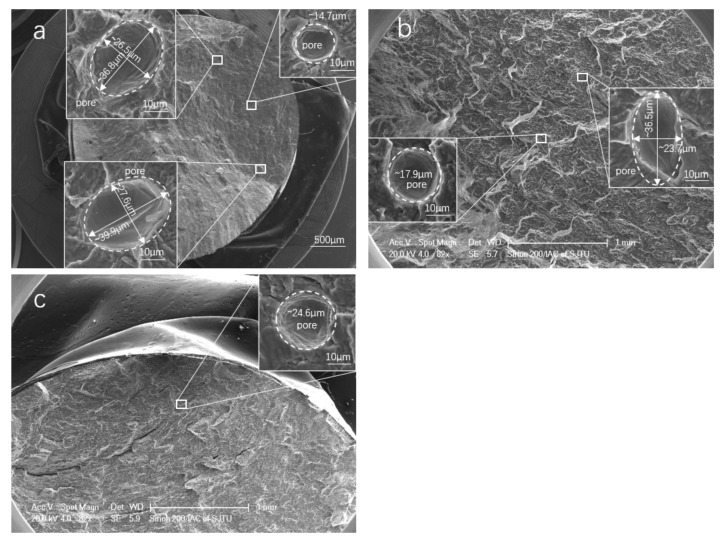
Internal defects in the fracture surface of LCF HT Ti-6Al-4V: (**a**) HT-0.8% and $N_{f}$ = 6686 cycles; (**b**) HT-1.2% and $N_{f}$ = 882 cycles; (**c**) HT-2.0% and $N_{f}$ = 116 cycles.

**Figure 17 materials-14-06276-f017:**
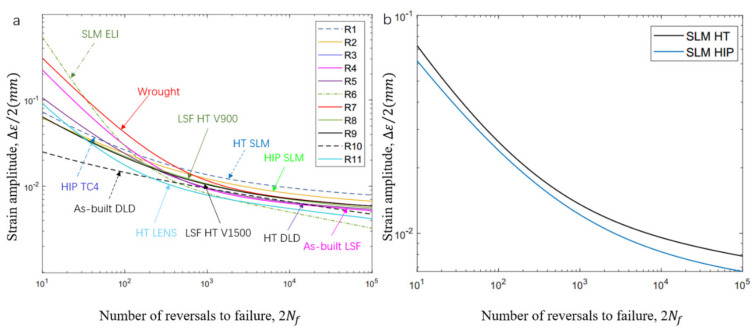
Comparison of LCF fitting curve with SLM HT- and HIP-treated Ti-6Al-4V: (**a**) LCF curve of various AM Ti-6Al-4V samples; (**b**) LCF properties of HT- and HIP-treated SLM samples.

**Figure 18 materials-14-06276-f018:**
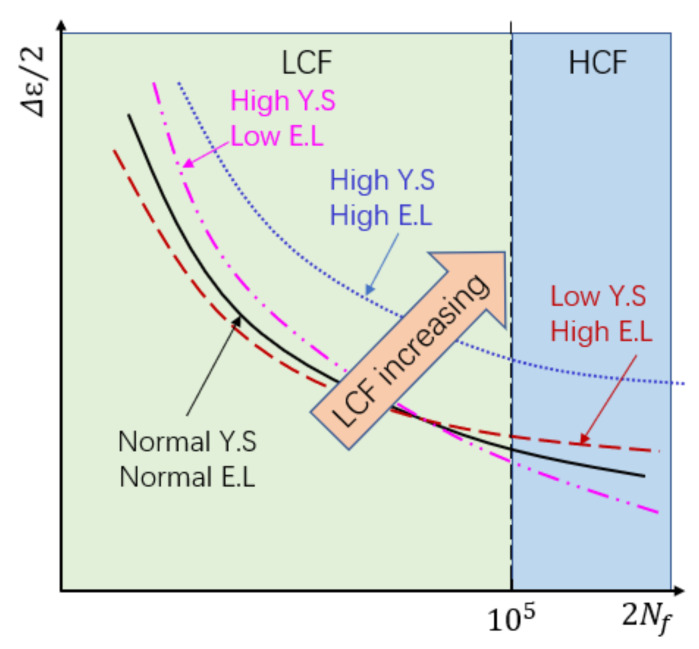
Illustration of the relationship among elongation to failure (EL), yield stress (YS), and LCF curve.

**Figure 19 materials-14-06276-f019:**
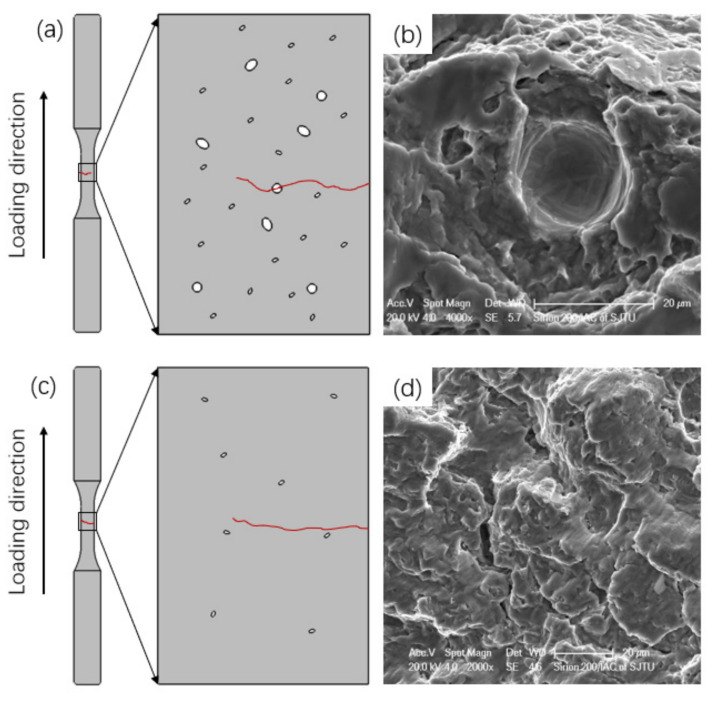
Schematic of internal defects along the fatigue crack propagation of SLM HT and HIP samples: (**a**) HT samples with pore defects; (**b**) internal defects on the fracture surface; (**c**) crack propagation of HIP sample; (**d**) typical region of fracture surface.

**Figure 20 materials-14-06276-f020:**
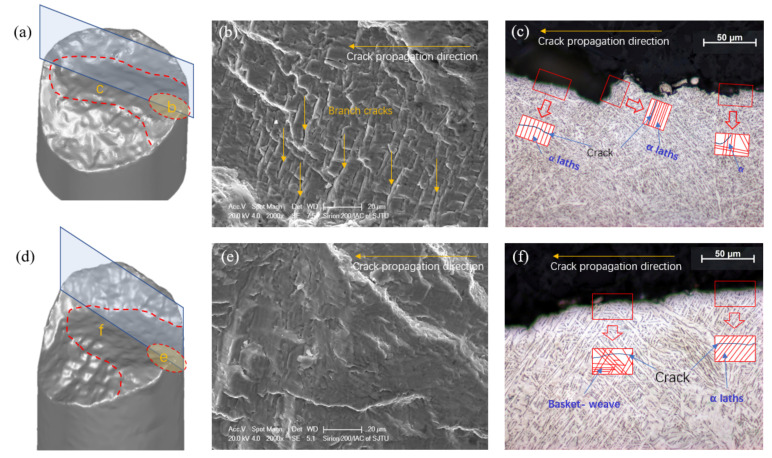
Fracture surfaces and crack propagation path: (**a**) fracture surface scanning model of HT sample at 0.8% strain amplitude; (**b**) crack initiation region of HT sample; (**c**) crack propagation path of HT sample; (**d**) fracture surface scanning model of HIP sample at 0.8% strain amplitude; (**e**) crack initiation region of HIP sample; (**f**) crack propagation path of HIP sample.

**Figure 21 materials-14-06276-f021:**
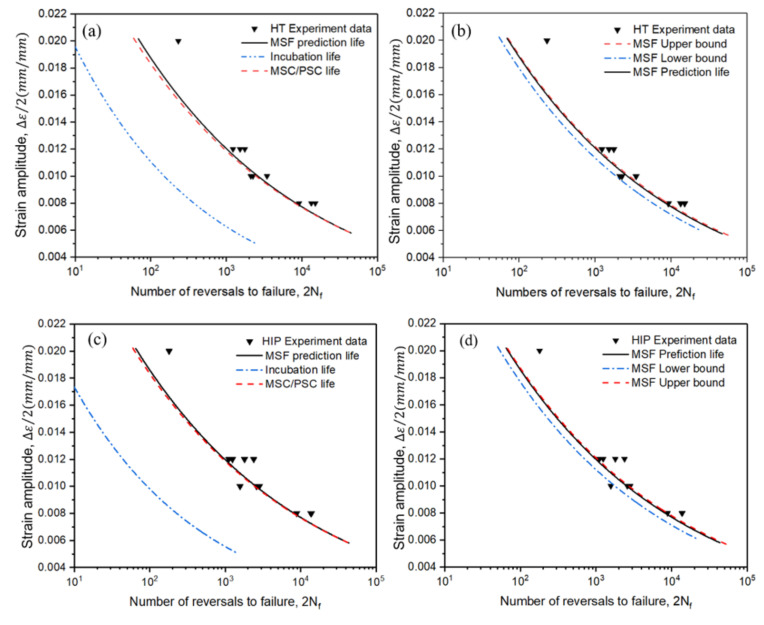
Fatigue life prediction using MSF model for HT and HIP samples: (**a**) crack incubation and small crack life for HT specimens; (**b**) lower bound and upper bound for HT specimens in MSF model; (**c**) crack incubation and small crack life for HIP-treated specimens; (**d**) lower bound and upper bound for HIP-treated specimens in MSF model.

**Table 1 materials-14-06276-t001:** Ti-6Al-4V powder chemical compositions used in this work (wt.%).

Al	Y	O	C	V	N	Fe	Ti
6.28	<0.001	0.074	0.012	3.97	0.012	0.131	Balance

**Table 2 materials-14-06276-t002:** List of fully reversed fatigue test parameters.

Δεa/2	f/Hz	Load Shape	Strain Ratio	Amount
0.008	0.313	Triangle	−1	3
0.010	0.250	Triangle	−1	3
0.012	0.208	Triangle	−1	3
0.020	0.125	Triangle	−1	1

**Table 3 materials-14-06276-t003:** Density and mechanical properties of HT- and HIP-treated SLM Ti-6Al-4V.

Properties	HT	HIP	Wrought
Density (g/cm^3^)	4.254	4.299	4.5
Elastic modulus E (GPa)	116.2	118.6	108.1
Yield stress, $\sigma_{y}$ (MPa)	964	913	904
Ultimate stress, $\sigma_{u}$ (MPa)	1115	1112	1078
Elongation to failure, T.E (%)	17.1	19	23.4
Cyclic modulus of elasticity, $E^{\prime }$ (GPa)	114.6	114.9	-
Fatigue strength coefficient, $\sigma_{f}^{\prime }$ (MPa)	1302	1076	-
Fatigue strength exponent, b	−0.0509	−0.0351	-
Fatigue ductility coefficient, $\varepsilon_{f}^{\prime }$	0.236	0.177	-
Fatigue ductility exponent, c	−0.582	−0.453	-
Cyclic strength coefficient, $\;K^{\prime }$ (MPa)	1311.6	1234.2	-
Cyclic strain hardening coefficient, $n^{\prime }$	0.0695	0.0674	-

**Table 4 materials-14-06276-t004:** Experiment results of strain-controlled fatigue test results for HT- and HIP-treated SLM Ti-6Al-4V.

$\varepsilon_{a}\;(\%)$	$\Delta \varepsilon_{p}/2\;(\%)$	$\Delta \varepsilon_{e}/2\;(\%)$	$\sigma_{a}\;(\mathrm{MPa})$	$\sigma_{m}\;(\mathrm{MPa})$	$2N_{f}\;(\mathrm{Reversals})$
**HT**	*-*	-	-	-	-
0.8	0.094	0.706	832	−16	13,372
0.8	0.074	0.726	763	−73	9194
0.8	0.094	0.706	825	−39	15,082
1.0	0.258	0.742	877	−27	3460
1.0	0.293	0.707	839	−34	2256
1.0	0.204	0.796	950	−42	2112
1.2	0.212	0.988	929	−39	1226
1.2	0.424	0.776	900	−10	1762
1.2	0.415	0.785	918	−29	1524
2.0	1.114	0.886	869	10	232
**HIP**	*-*	-	-	-	-
0.8	0.129	0.671	786	−10	13,636
0.8	0.129	0.671	787	−31	8860
0.8	0.141	0.659	781	−16	13,872
1.0	0.299	0.701	831	−26	2836
1.0	0.293	0.707	836	−35	2598
1.0	0.245	0.755	849	−45	1572
1.2	0.489	0.711	834	10	1800
1.2	0.476	0.724	845	−28	1114
1.2	0.431	0.769	840	2	1244
2.0	1.246	0.754	939	−36	180

**Table 5 materials-14-06276-t005:** The observed defects in fracture surface of LCF samples.

No.	Specimen	Sizes of Defects (μm)	Location of Defects	Cycles to Failure
1	HT-5#-0.8%	14.7, (26.5, 36.8), (27.6, 39.9)	interior	6686
2	HT-3#-1.2%	17.9, (23.7, 36.5)	interior	882
3	HT-8#-2.0%	24.6	interior	116

Note: #The diameter of defect size in brackets indicates the lengths of the minor and major axes of the elliptical defects.

**Table 6 materials-14-06276-t006:** Low cycle fatigue $\Delta \varepsilon /2–2N_{f}$ fitting curves of AM Ti-6Al-4V and wrought samples from the literature.

Number	Process	Yield Stress (MPa)	Elongation (%)	LCF Properties				Reference
				${\sigma^{\prime }}_{f}$	b	${\varepsilon^{\prime }}_{f}$	**c**	
1	HT SLM	964	17.1	0.01366	−0.05085	0.23615	−0.5915	This work
2	HIP SLM	913	19	0.009358	−0.03511	0.17677	−0.5208	
3	HT lens	959	3.7	0.015	−0.111	0.736	−0.967	[37]
4	As-built	893	11	0.01177	−0.07162	2.13535	−1.0007	[38]
5	HIP LSF	872	12.3	0.1028	−0.0575	0.5899	−0.78261	[39]
6	SLM Ti-6Al-4V ELI	1015	10	0.02761	−0.186	15.35	−1.47	[40]
7	Wrought	>825	>10	0.013	−0.07	2.69	−0.96	[41]
8	HT LSF	791.6	18.2	0.0097	−0.05217	0.20621	−0.57527	[23]
9	HT LSF	839.5	17.8	0.00946	−0.04474	0.21957	−0.60018	
10	As-built DLD	908	3.8	0.022	−0.135	0.03	−0.53	[42]
11	HT DLD	957	3.4	0.015	−0.111	0.736	−0.967	

**Table 7 materials-14-06276-t007:** Parameters used in the MSF model for SLM HT and HIP Ti-6Al-4V alloy.

	Coefficients	Value	Ref.
Crack Incubation	$C_{n}$	HT:0.01366	-
		HIP:0.00935	-
	$C_{m}$	HT:0.236	-
		HIP:0.177	-
	α	HT:−0.5915	-
		HIP:−0.5208	-
	q	2.45	[22]
	$y_{1}$	100	[21]
	r	0.2	-
	$\eta_{lim}$	0.3	-
	$\varepsilon_{per}$	HT: 0.35% HIP: 0.36%	-
	$\varepsilon_{th}$	0.14%	-
Small crack (MSC/PSC)	θ	0	-
	χ	0.26	[51]
	$C_{Ι}$	$1.2\times 10^{5}$	-
	$C_{\mathrm{II}}$	0.25	-
	n	4.6	-
	$f\left(\bar{\varphi }\right)$	~1	-
	$\Delta CTD_{th}$	$2.95\times 10^{-4}\mu m$	[51]

## Data Availability

The data presented in this study are available on request from the corresponding authors.

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
