# Peer review of "The Low-Cycle Fatigue Behavior, Failure Mechanism and Prediction of SLM Ti-6Al-4V Alloy with Different Heat Treatment Methods"

_materials, 2021, doi:10.3390/ma14216276_

Round 1
Reviewer 1 Report
Good work, testing and discussion are OK
The microstructure of the LPBF-ed components depends on three main aspects: the solidification mode, LSS and volumetric energy density (VED) Moreover, these features depend on the following specific parameters: the laser power (P), layer thickness (t), hatching space (h) and laser speed (V_b).
- The epitaxial growth of crystals is frequently reported in literature as the most important phenomenon governing the columnar grain microstructure and causing it to appear in nearly all printed alloys, such as Inconel 718 Several studies on the solidification of metals during LPBF, welding or casting processes agree that the thermal gradient (G ⃑) and solidification rate (V_i) are the most significant factors that govern the columnar grain growth. In the case of Inconel, it can be concluded that grains are columnar at essentially any value of the solidification rate for a sufficiently high thermal gradient. Competitive grain growth frequently occurs in the LPBF process, which implies that the prediction of texture is more complex and closely related to the manufacturing parameters, especially the energy density and laser scanning strategies. You must discuss it along with your results.
- People are making great advances, in many projects, I read the minimum size of structures in: Materials 2021, 14(7), 1588; https://doi.org/10.3390/ma14071588. But they did not tested the probes.
- Make a better discussion. Figure 2 is about heat treatments, di you use HIP in some samples, why not?
Figure 5 is a good opportunity to discuss above concepts.
Look for works in Materials above and others.
Author Response
Review #1: Good work, testing and discussion are OK.
- The microstructure of the LPBF-ed components depends on three main aspects: the solidification mode, LSS and volumetric energy density (VED) Moreover, these features depend on the following specific parameters: the laser power (P), layer thickness (t), hatching space (h) and laser speed (V_b).The epitaxial growth of crystals is frequently reported in literature as the most important phenomenon governing the columnar grain microstructure and causing it to appear in nearly all printed alloys, such as Inconel 718 Several studies on the solidification of metals during LPBF, welding or casting processes agree that the thermal gradient (G ⃑) and solidification rate (V_i) are the most significant factors that govern the columnar grain growth. In the case of Inconel, it can be concluded that grains are columnar at essentially any value of the solidification rate for a sufficiently high thermal gradient. Competitive grain growth frequently occurs in the LPBF process, which implies that the prediction of texture is more complex and closely related to the manufacturing parameters, especially the energy density and laser scanning strategies. You must discuss it along with your results.
We thank the Reviewer for the description and explanation of the additive manufacturing technology during the process and the critical parameters in the process. The Reviewer also pointed out the microstructure evolution phenomenon and the most significant factors that govern the columnar growth. We also want to thanks for your recommendation, the paper has a great work, we read it and learn quite useful results from it. It proposed process parameters for the fabrication of ultralight components. We made the change and improvement of the introduction chapter.
On page 1 and 2:
With carefully chosen process parameters, SLM titanium alloy have possibility to achieve near fully dense parts [7]. Parts with comparable mechanical properties to those of traditionally fabricated titanium alloy has been reported [11]- [12]. Tt is evident that, to produce near fully dense products, the key process parameters concern the laser power density, layer thickness, sintering rate, the manufacturing strategy [8]- [9], which are all about the mechanism of densification during the manufacturing process [9].
In SLM, due to the high cooling rate, there is an anisotropic α^' martensitic phase which are contain within prior-β grains oriented epitaxially in a perpendicular direction to the layers [4]. This typical microstructure alloys have high strength but with poor ductility as well as certain anisotropy [10]. Therefore, to achieve a balanced property between strength and ductility, post treatment involving heat treatment have been introduced [11].
We also want to state that the emphasis of this work is to investigate the fracture mechanism of SLM parts at different heat treatment methods. Therefore, wo focused on the heat treatment effects and other process parameters were not the critical factors we need to discuss.
- People are making great advances, in many projects, I read the minimum size of structures in: Materials 2021, 14(7), 1588; https://doi.org/10.3390/ma14071588. But they did not tested the probes.
Thanks for your recommendation, the paper has a great work, we read it and learn quite useful results from it. It proposed process parameters for the fabrication of ultralight components. We have added the mentioned reference shown as follows, on page 27:
[8] Calleja‐ochoa A, Gonzalez‐barrio H, de Lacalle NL, Martínez S, Albizuri J, Lamikiz A. A new approach in the design of microstructured ultralight components to achieve maximum functional performance, 2021
- Make a better discussion. Figure 2 is about heat treatments, di you use HIP in some samples, why not?
We did the discussion of the heat treatments in this section. The heat treatment methods applied in this work were heat treatment and hot isostatic pressing. The samples in this work were divided into two parts, the half of the specimens are HIPed to investigate the effects of heat treatment on fatigue properties.
- Figure 5 is a good opportunity to discuss above concepts.
In section 3.1 the 3D map of grain morphology was shown in Figure.5. We discussed the microstructure patterns of the samples undergone the two heat treatment methods. It can be inferred that the same process parameters such as laser power, layer thickness, sintering rate led to similar microstructure pattern.
- Look for works in Materials above and others.
Yes, of course we will. Except the reference literature the Reviewer recommend, we also looked for more works on fabrication parameters and their effects on the properties in Materials and learn from the, we also added the following paper published in Materials on page 27:
[2] Leonhard Hitzler,* Markus Merkel, Wayne Hall and AO. A Review of Metal Fabricated with Laser- and Pow-der-Bed Based Additive Manufacturing Techniques:Process,Nomenclature, Materials, Achievable Properties, and its Utilization in the Medical Sector, 2018
Reviewer 2 Report
Very well, written ms.
Introduction: last paragraph can be either omitted or relocated.Please state hypothesys of your study.
Materials and methods:
write in the past tense
you mascinned the printed specimes to final form for tensile testing. Did it affect the properties. Please state
Provide number of specimens for each group
Author Response
Review #2: Very well, written ms.
- The last paragraph can be either omitted or relocated. Please state hypothesis of your study.
Thanks for your suggestions which are useful for the improvement of the manuscript.
The last paragraph has been rewritten and part of the content was relocated and rewritten on page 2.
This study investigated the effects of HIP and standard HT methods on the low cycle fatigue properties of SLM parts. To eliminate the surface roughness effects, which are the most detrimental factor for fatigue property, all the surfaces of specimens in this study were machined with identical fine surface finishing. Low cycle fatigue (LCF) and tensile test of HT and HIP treated samples were carried out to investigate the mechanical properties behaviour and LCF performance. The relationship of mechanical properties and LCF properties were discussed and summarized. In addition, this research aimed at predicting fatigue life of HT and HIP specimens using the multi-stage fatigue (MSF) model proposed by McDowell et al. [21] and calibrate the results in AM Ti-6Al-4V alloys.
The hypothesis in this study is the relationship between the tensile properties and LCF properties and the LCF trend can be predicted by the yield stress and ductility. In addition, difference of the two different heat treatment methods on the fracture mechanism were investigated and the MSF model can be applied in this kind of SLM Ti-6Al-4V material and predicting results has a good agreement with the experiment results.
- Materials and methods: write in the past tense.
The tense was revised to the past tense.
- you machined the printed specimens to final form for tensile testing. Did it affect the properties? Please state.
The aim of this study is to investigate the effect of HT and HIP treatment on the SLM Titanium alloy. We assume the components in this study will be applied in the aerospace or medical field, where the surface of parts should be machined and polished in real application. Nevertheless, surface machined samples were not superior to as-built samples in tensile strength but have much higher fatigue life comparing to the AB samples.
- Provide number of specimens for each group
The number of specimens for each group can be seen in Table 4 on page 10. There were three specimens performed at the 0.8%, 1.0% and 1.2% strain amplitude, and one specimen at 2.0% strain level for both HT and HIP condition.
Table 4. Experiment results of strain-controlled fatigue test results for HT and HIP treated SLM Ti-6Al-4V
(%) |
(%) |
(%) |
(MPa) |
(MPa) |
2(Reversals) |
HT |
|||||
0.8 |
0.094 |
0.706 |
832 |
-16 |
13372 |
0.8 |
0.074 |
0.726 |
763 |
-73 |
9194 |
0.8 |
0.094 |
0.706 |
825 |
-39 |
15082 |
1.0 |
0.258 |
0.742 |
877 |
-27 |
3460 |
1.0 |
0.293 |
0.707 |
839 |
-34 |
2256 |
1.0 |
0.204 |
0.796 |
950 |
-42 |
2112 |
1.2 |
0.212 |
0.988 |
929 |
-39 |
1226 |
1.2 |
0.424 |
0.776 |
900 |
-10 |
1762 |
1.2 |
0.415 |
0.785 |
918 |
-29 |
1524 |
2.0 |
1.114 |
0.886 |
869 |
10 |
232 |
HIP |
|||||
0.8 |
0.129 |
0.671 |
786 |
-10 |
13636 |
0.8 |
0.129 |
0.671 |
787 |
-31 |
8860 |
0.8 |
0.141 |
0.659 |
781 |
-16 |
13872 |
1.0 |
0.299 |
0.701 |
831 |
-26 |
2836 |
1.0 |
0.293 |
0.707 |
836 |
-35 |
2598 |
1.0 |
0.245 |
0.755 |
849 |
-45 |
1572 |
1.2 |
0.489 |
0.711 |
834 |
10 |
1800 |
1.2 |
0.476 |
0.724 |
845 |
-28 |
1114 |
1.2 |
0.431 |
0.769 |
840 |
2 |
1244 |
2.0 |
1.246 |
0.754 |
939 |
-36 |
180 |
Reviewer 3 Report
The paper gives a structured overview of the LCF behaviour of Ti-6Al-4V SLM parts but has some formal weaknesses. The following general points need to be revised:
- In addition to the spell check, the references of the figures and tables need to be checked and corrected. Furthermore, the font size needs to be harmonized especially in chapter 4.
- The description of the figures needs to be checked especially for figures 7 and 8.
- The legends and descriptions in the figures are difficult to read and should be revised accordingly.
Furthermore the following specific points need to be revised:
- Is there an author missing?
- Figure 1: Why does the chosen scanning strategy have a 90° rotation instead of a 67° rotation of the layers?
- 5 line 176: which Zeiss microscope and which SEM machine have been used?
Author Response
Review #3:response attached

Reviewer 4 Report
Dear authors, while the topic is of high interest, the manuscript does need substantial improvements.
Foremost, please double-check and correct the terminology according to the current standard, e.g. SLM is a sub-process of powder-bed fusion. For your convenience, please see the ISO 52900 standard on Additive Manufacturing-General Principles-Terminology, Geneva, Switzerland 2015, and/or a recent review article addressing the nomenclature of AM processes (doi:10.1002/adem.201700658)
Page 1 Line 40 and following: “Many manufacturing parameters” is a rather vague description. This is the followed by the term of “process parameters” and it remains unclear whether or not you mean the same parameters or have a differing classification of manufacturing vs. process parameters. However, the powder itself, which is named just after, is neither! It is the feedstock or raw material used and not a parameter. Please be more thorough in the wording and description to avoid confusion of readers which are not experts in this area and may not know the differences.
Page 2 line 58 and following: It is described that the intrinsic limitations lead to porosity, surface roughness and residual stresses. Since these intrinsic limitations based on your definition are limited to the thermal history the part or sample experiences, there is, based on this definition, only a weak connection to the surface roughness –which major dependencies are layer thickness, inclination angle, and the size of stuck powder particles – and the porosity, which requires the process to be within the stable process window to avoid lack of fusion and keyhole porosity. So, while the named factors do play a role, they are by no means the only ones and the direct link presented cannot be made in this manner.
Just after surface roughness is named as the key factor limiting fatigue performance, even more important than internal defects. However, then HIP is presented as a possible solution, which does not improve the surface roughness, which based on your own argumentation is the key factor. This chain of argumentation is in itself inconsistent.
It is recommended to include AB tests to emphasise the arguments made about HT and HIP superiority.
Density measurements should be included to show how much improvements were achieved via HIP. This will allow to further quantify the seen alterations and how much of it is due to reduced porosity or simply via heat treatment at higher temperature.
Formalities:
- There appears to be either one author missing or the and in the author line needs to be shifted before K. Nikbin
- Variables and parameters should be written in italic, whereas units are non-italic
- Use a space between number and unit (mixed throughout the manuscript)
Round 2
Reviewer 4 Report
To the original comments:
1) Foremost, please double-check and correct the terminology according to the current standard, e.g. SLM is a sub-process of powder-bed fusion. For your convenience, please see the ISO 52900 standard on Additive Manufacturing-General Principles-Terminology, Geneva, Switzerland 2015, and/or a recent review article addressing the nomenclature of AM processes (doi:10.1002/adem.201700658)
To page 1 line 38:
Description of the PBF process is still inaccurate. PBF does not necessitate a full melting of the powder!! For example, indirect laser sintering, which uses polymer coated powder particles and hence, does not fully melt the feed stock still falls within the PBF category! Please correct the terminology.
6) Density measurements should be included to show how much improvements were achieved via HIP. This will allow to further quantify the seen alterations and how much of it is due to reduced porosity or simply via heat treatment at higher temperature.
The density of the samples was measured after fabrication. The results showed that the carefully chosen process parameters can help to build samples with very close density to the conventional manufactured materials. The difference between the HT and HIP samples were unnoticeable, this can be verified in the section 3.4 table 6 on page 18.
In this instance, point 4 in your conclusion is misleading, as it clearly suggests that HIP is superior due to closure of voids, i.e. a reduced porosity. Please be consistent in your argumentation.
Formalities:
Page 1 line 27: the abbreviation AM is stated twice, once in brackets and once in the text
Author Response
Response to Editors and Reviewers
The authors would like to thank the reviewers for their thorough work and helpful comments. All of these comments and suggestions has been addressed and integrated in the final manuscript.
Changes made to the manuscript have been highlighted in red in this document. Comments made by the reviewers are shown in black. Our response to comments is shown in blue.
Review #4: To the original comments.
1. To page 1 line 38:Description of the PBF process is still inaccurate. PBF does not necessitate a full melting of the powder!! For example, indirect laser sintering, which uses polymer coated powder particles and hence, does not fully melt the feed stock still falls within the PBF category! Please correct the terminology.
We thank the Reviewer for the description and explanation of the PBF process technology. These suggestions are very help for the authors to understand the terminology and to improve this study.
The description of the PBF technology has been rewritten by more understanding the paper recommended by the reviewer and show as bellow:
On page 1 from line 38-44.
“Powder bed fusion technology (PBF) is a specific developed subset of AM technologies which use a concentrated energy beam to melt powder bed-based of polymer, metal or ceramic based raw materials layer-by-layer. Moreover, PBF processes vary on base of the type of applied power source, for instance, laser or electron beam. Selective laser melting (SLM), utilizing the laser beam as energy source is one of most widely used PBF process [2] and appealing for the fabrication of Ti-6Al-4V”
2. In this instance, point 4 in your conclusion is misleading, as it clearly suggests that HIP is superior due to closure of voids, i.e. a reduced porosity. Please be consistent in your argumentation..
The suggestion of the reviewer is very convincing that the evidence is necessary to support the conclusion point 4. To obtain the density, we measured the density of the samples and exhibited the average result in Table 3 and Table 3 is revised as follows.
Table 3. Density and mechanical properties of the HT and HIP treated SLM Ti-6Al-4V.
Properties |
HT |
HIP |
Wrought |
|||
Density (g/cm3) |
|
|
4.254 |
4.299 |
4.5 |
|
Elastic Modulus (GPa) |
116.2 |
118.6 |
108.1 |
|||
Yield Stress, (MPa) |
964 |
913 |
904 |
|||
Ultimate stress, (MPa) |
|
1115 |
1112 |
1078 |
||
Elongation to Failure, (%) |
|
17.1 |
19 |
23.4 |
||
|
|
|
|
|
|
|
Cyclic Modulus of Elasticity, (GPa) |
114.6 |
114.9 |
|
|||
Fatigue Strength Coefficient, (MPa) |
1302 |
1076 |
|
|||
Fatigue Strength Exponent, |
|
-0.0509 |
-0.0351 |
|
||
Fatigue Ductility Coefficient, |
0.236 |
0.177 |
|
|||
Fatigue Ductility Exponent, |
|
-0.582 |
-0.453 |
|
||
Cyclic Strength Coefficient, (MPa) |
|
1311.6 |
1234.2 |
|
||
Cyclic Strain Hardening Coefficient, |
|
0.0695 |
0.0674 |
|
||
|
|
|
|
|
|
|
We also added a description in page 9 line 301:
In addition, to verify the effect of HIP treatment on closing the internal defects, the density of the samples was measured and listed in Table 3.
3. Formalities:
Page 1 line 27: the abbreviation AM is stated twice, once in brackets and once in the text
The abbreviation AM in page 1 line 27 has been removed.
We thank the Reviewer for the suggestions on all the technical and language issues. We have carefully revised the manuscript again for correcting all the issues and highlighted the corrections by tracking the changes.
Yours Sincerely!
The Authors